# CHATPAINTER: IMPROVING TEXT TO IMAGE GENERATION USING DIALOGUE

**Shikhar Sharma**
Microsoft Research

**Dendi Suhubdy**
Université de Montréal, MILA

**Vincent Michalski**
Université de Montréal, MILA*

**Samira Ebrahimi Kahou**
Microsoft Research

**Yoshua Bengio**
Université de Montréal, MILA

## ABSTRACT

Synthesizing realistic images from text descriptions on a dataset like Microsoft Common Objects in Context (COCO), where each image can contain several objects, is a challenging task. Prior work has used text captions to generate images. However, captions might not be informative enough to capture the entire image and insufficient for the model to be able to understand which objects in the images correspond to which words in the captions. We show that adding a dialogue that further describes the scene leads to significant improvement in the inception score and in the quality of generated images on the COCO dataset.

## 1 MOTIVATION

Automatic generation of realistic images from text descriptions has numerous potential applications, for instance in image editing, in computer gaming or in law enforcement. Spurred by the recent successes of Variational Autoencoders (VAEs) (Kingma & Welling, 2014; Van Den Oord et al., 2016; Cai et al., 2017) and Generative Adversial Networks (GANs) (Goodfellow et al., 2014; Arjovsky et al., 2017; Gulrajani et al., 2017; Karras et al., 2018), there has been a lot of recent work and interest in the research community on image generation from text captions (Mansimov et al., 2016; Reed et al., 2016a; Zhang et al., 2017). Current state-of-the-art models are capable of generating realistic images on datasets of birds, flowers, room interiors, faces etc., but don't do very well on datasets like COCO (Lin et al., 2014) which contain several objects within a single image and where subjects are not always centred in the image. A caption for an image of a flower can usually describe most of the relevant details of the flower (e.g. "This flower has overlapping pink pointed petals surrounding a ring of short yellow filaments"). However, for the COCO dataset a caption might not contain all the relevant details about the foreground and the background. Due to this complexity and limited amount of image-caption paired data, the model might not always be able to understand which objects in the image correspond to which words in the caption. Previous work in the literature has found that conditioning on auxiliary data such as category labels (Mirza & Osindero, 2014), or object location and scale (Reed et al., 2016b) helps in improving the quality of generated images.

Sketch artists typically have a back and forth conversation with witnesses when they have to draw a person's sketch, where the artist asks for more details and refines the sketch while the witness provides requested details and feedback on the current state of the sketch. We hypothesize that conditioning on a similar conversation about a scene in addition to a caption would significantly improve the generated image's quality and we explore this idea in this paper. For this, we pair captions provided with the COCO dataset with dialogues from the Visual Dialog dataset (VisDial) (Das et al., 2017). These dialogues were collected using a chat interface pairing two workers on Amazon Mechanical Turk (AMT). One of them was assigned the role of an 'answerer', who could see a COCO image together with its caption and had to answer questions about that image. The other was assigned the role of the 'questioner' and could see only the image's caption. The questioner had to ask questions to be able to imagine the scene more clearly. For more details about the dataset, consult Appendix A. Though we just demonstrate improvements over the StackGAN (Zhang et al., 2017) model in this paper, this additional dialogue module can be added to any caption-to-image-generation model and is an orthogonal contribution.

---

*Part of this work was completed while the author was an intern at Microsoft Research Montréal.

## 2 MODEL

(a) Stage-I model

(b) Stage-II model

Figure 1: ChatPainter: (a) Stage-I of the model generates a $64 \times 64$ image conditioned on a caption and the corresponding dialogue. (b) Stage-II of the model generates a $256 \times 256$ image conditioned on Stage-I's $64 \times 64$ generated image and the caption and corresponding dialogue.

Our model, ChatPainter, builds upon the StackGAN model and is shown in Figure 1. We generate caption embedding $\varphi_t$ by encoding the captions with a pre-trained encoder (Reed et al., 2016a). We generate dialogue embeddings $\zeta_d$ by two methods: (a) **Non-recurrent encoder**: We collapse the entire dialogue into a single string and encode it with a pre-trained Skip-Thought (Kiros et al., 2015) encoder. (b) **Recurrent encoder**: We generate Skip-Thought vectors for each turn of the dialogue then encode them with a bidirectional LSTM-RNN. We then concatenate the caption and dialogue embeddings and this is passed as input to the Conditioning Augmentation (CA) module. As in Stack-GAN, the CA module produces latent variable inputs for the generator from the embeddings. We also adapt their regularization term to encourage smoothness over the conditioning manifold for our dialogue embeddings: $D_{KL}(\mathcal{N}(\mu(\varphi_t, \zeta_d), \mathrm{diag}(\sigma(\varphi_t, \zeta_d)))||\mathcal{N}(0, I))$, where $D_{KL}$ is the Kullback-Leibler divergence. In the CA module, a fully connected layer is applied over the input that generates $\mu$ and $\sigma$ which are both $N_g$ dimensional. The module samples $\epsilon$ from $\mathcal{N}(0, I)$. Finally, the conditioning variables $\hat{c}$ are computed as $\hat{c} = \mu + \sigma \odot \epsilon$, where $\odot$ is the element-wise multiplication operator. Thus, the conditioning variables $\hat{c}$ are effectively samples from $\mathcal{N}(\mu(\varphi_t, \zeta_d), \mathrm{diag}(\sigma(\varphi_t, \zeta_d)))$.

**Stage-I** The conditioning variables for Stage-I, $\hat{c}_0$, are concatenated with $N_z$-dimensional noise, $z$, drawn from a random normal distribution, $p_z$. The Stage-I generator upsamples this input representation to a $W_0 \times H_0$ image. This Stage-I image is expected to be blurry and a rough version of the final one. The discriminator downsamples this image to $M_d \times M_d \times N_{di}$. $\hat{c}_0$ is then spatially replicated to $M_d \times M_d \times N_d$ and concatenated with the downsampled representation. This is further downsampled to a scalar value between 0 and 1. The model is trained by alternating between maximizing $\mathcal{L}_{D_0}$ and minimizing $\mathcal{L}_{G_0}$:

$$\mathcal{L}_{D_0} = \mathop{\mathbb{E}}_{(I_0, t, d) \sim p_{data}} [\log D_0(I_0, \varphi_t, \zeta_d)] + \mathop{\mathbb{E}}_{z \sim p_z, (t, d) \sim p_{data}} [\log(1 - D_0(G_0(z, \hat{c}_0), \varphi_t, \zeta_d))],$$

$$\mathcal{L}_{G_0} = \mathop{\mathbb{E}}_{z \sim p_z, (t, d) \sim p_{data}} [\log(1 - D_0(G_0(z, \hat{c}_0), \varphi_t, \zeta_d))] + \lambda D_{KL}(\mathcal{N}(\mu(\varphi_t, \zeta_d), \mathrm{diag}(\sigma(\varphi_t, \zeta_d)))||\mathcal{N}(0, I)),$$

where $I_0$ is the real image, $t$ is the text caption, $d$ is the dialogue, $p_{data}$ is the true data distribution, $\lambda$ is the regularization coefficient, $G_0$ is the Stage-I generator, and $D_0$ is the Stage-I discriminator.

**Stage-II** The Stage-II generator, $G$, first downsamples generated stage-I images to $M_g \times M_g \times N_{gi}$. The conditioning variables for Stage-II, $\hat{c}$, are generated and then spatially replicated to $M_g \times M_g \times N_g$ and finally concatenated to the downsampled image representation. For Stage-II training, in case of the recurrent dialogue encoder, the RNN weights are copied from Stage-I and kept fixed. The concatenated input is passed through a series of residual blocks and is then upsampled to a $W \times D$ image. The Stage-II discriminator, $D$, downsamples the input image to $M_d \times M_d \times N_{di}$. $\hat{c}$ is then spatially replicated to $M_d \times M_d \times N_d$ and concatenated with the downsampled representation which is further downsampled to a scalar value between 0 and 1. The Stage-II model is trained by alternating between maximizing $\mathcal{L}_D$ and minimizing $\mathcal{L}_G$:

$$\mathcal{L}_D = \mathop{\mathbb{E}}_{(I, t, d) \sim p_{data}} [\log D(I, \varphi_t, \zeta_d)] + \mathop{\mathbb{E}}_{s_0 \sim p_{G_0}, (t, d) \sim p_{data}} [\log(1 - D(G(s_0, \hat{c}), \varphi_t, \zeta_d))],$$

$$\mathcal{L}_G = \mathop{\mathbb{E}}_{s_0 \sim p_{G_0}, (t, d) \sim p_{data}} [\log(1 - D(G(s_0, \hat{c}), \varphi_t, \zeta_d))] + \lambda D_{KL}(\mathcal{N}(\mu(\varphi_t, \zeta_d), \mathrm{diag}(\sigma(\varphi_t, \zeta_d)))||\mathcal{N}(0, I)),$$

where $I$ is the real image, and $s_0$ is the image generated from Stage-I. In our experiments, $N_z = 100$, $W_0 = 64$, $H_0 = 64$, $M_d = 4$, $N_{di} = 512$, $N_d = 128$, $M_g = 16$, $N_{gi} = 512$, $N_g = 128$, $W = 256$, $D = 256$, and $\lambda = 2$ – same as that in the StackGAN model. The architecture of the upsample, downsample and residual blocks is kept the same as that in the original StackGAN. For further training details, refer to Appendix B.

# 3 RESULTS

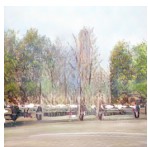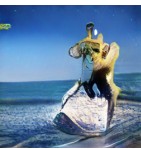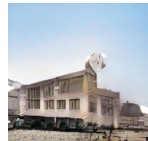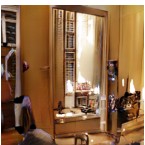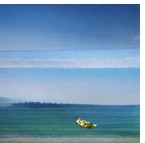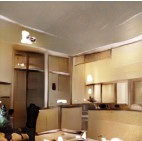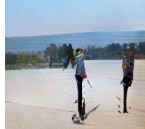

Figure 2: Example $256 \times 256$ images generated by our recurrent encoder ChatPainter model on the COCO test set. Best viewed in color. Images are cherry-picked from a larger random sample.

Table 1: Inception scores for generated images on the COCO test set.[1]

| Model | Inception Score |
|---|---|
| Reed et al. (2016a) | $7.88 \pm 0.07$ |
| StackGAN (Zhang et al., 2017) | $8.45 \pm 0.03$ |
| ChatPainter (non-recurrent) | $9.43 \pm 0.04$ |
| ChatPainter (recurrent) | $\mathbf{9.74 \pm 0.02}$ |
| Hong et al. (2018) | $11.46 \pm 0.09$ |
| AttnGAN (Xu et al., 2017) | $\mathbf{25.89 \pm 0.47}$ |

We present some of the more realistic images generated by our recurrent encoder ChatPainter in Figure 2. We report inception scores (Salimans et al., 2016) on the images generated from our models in Table 1 and compare with other recent models. We see that the ChatPainter model, which is conditioned on additional dialogue information, gets higher inception score than the StackGAN model just conditioned on captions. Also, the recurrent version of ChatPainter gets higher inception score than the non-recurrent version. This is likely due to it learning better encoding of the dialogues as the Skip-Thought encoder isn't trained with very long sentences, which is the case in the non-recurrent version as we collapse the dialogue in a single string. The inception score evaluation methodology is described in Appendix C and a larger sample of the generated images is presented in Appendix D.

# 4 DISCUSSION AND FUTURE WORK

In this paper, apart from conditioning on image captions, we additionally conditioned the ChatPainter model on publicly available dialogue data and obtained significant improvement in inception score on the COCO dataset. While many of the generated $256 \times 256$ images look quite realistic, the StackGAN family of models (including ChatPainter) has several limitations and exhibits some of the issues other GANs also suffer from. The StackGAN family is able to generate photo-realistic images easily on restricted-domain datasets such as those on flowers and birds but on COCO, it is able to generate images that exhibit strong global consistency but do not produce recognizable objects in many cases. The current training loss formulation also makes it susceptible to *mode collapse*. Training the model with dialogue data is also not very stable. Recent improvements in the literature such as training with the WGAN-GP loss can help mitigate these issues to some extent. Using an auxiliary loss for the discriminator by doing object recognition or caption generation from the generated image should also lead to improvements as has been observed in prior work on other image generation tasks. The non-end-to-end-training also leads to longer training time and loss of information which can be improved upon by growing the model progressively layer-by-layer as done by Karras et al. (2018). An interesting research direction we wish to explore further is to generate an image at each turn of the conversation (or modify the previous time-step's image) using dialogues as a feedback mechanism. In the sketch-artist scenario discussed in Section 1, the sketch artist would make several changes to the image as the conversation progresses and the future conversation also would depend on the image at that point in the conversation, However, no such publicly available data exists yet to the best of our knowledge and we plan to collect such a dataset soon. Image generation guided by dialogue has tremendous potential in the areas of image editing, computer gaming, digital art, law enforcement, etc., and is a promising future research direction in our opinion.

---

[1]The two best-performing methods were released while writing this manuscript and we will evaluate the effect of our scheme using these methods as base architecture in future work.

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

## A  DATASETS

Table 2: An example of the input data, the corresponding dataset image, and the corresponding image generated by our best ChatPainter model.

| Input | Dataset image | Generated image |
|---|---|---|
| Caption: adult woman with yellow surfboard standing in water. 
 Q: is the woman standing on the board?  A: no she is beside it. 
 Q: how much of her is in the water?    A: up to her midsection. 
 Q: what color is the board?                              A: yellow. 
 Q: is she wearing sunglasses?                              A: no. 
 Q: what about a wetsuit?          A: no she has on a bikini top. 
 Q: what color is the top?                A: orange and white. 
 Q: can you see any other surfers?                       A: no. 
 Q: is it sunny?      A: the sky isn't visible but it appears to be a nice day. 
 Q: can you see any palm trees?                          A: no. 
 Q: what about mountains?                                A: no. |  |  |

In our experiments, we used images and their captions from the COCO dataset. COCO covers 91 categories of objects, grouped into 11 super-categories of objects such as *person and accessory*, *animal*, *vehicle*, etc. We use the '2014 Train' set as our training set and the '2014 Val' set as our test set. The train set consists of $\sim 80K$ images and includes five captions for each image. The test set consists of $\sim 40K$ images along with their captions.

We obtain dialogues for these images from the VisDial dataset. VisDial consists of 10 question-answer conversation turns per dialogue and has one dialogue for each of the COCO images. VisDial was collected by pairing two crowd-workers and having them talk about an image as described in Section 1. Hence, we have $\sim 80K$ dialogues for the training set and $\sim 40K$ for the test set.

Table 2 shows the corresponding caption, dialogue inputs, and the test set image for an image generated by our best ChatPainter model.

## B  TRAINING DETAILS

Similar to StackGAN, we use a matching-aware discriminator (Reed et al., 2016a), that is trained using "real" pairs consisting of a real image together with matching caption and dialogue, and "fake" pairs that consist either of a real image together with another images's caption and dialogue or a generated image with the corresponding caption and dialogue. We train both stages for $800$ epochs using the Adam optimizer (Kingma & Ba, 2014). The initial learning rate for all experiments is $0.0002$. We decay the learning rate to half of its previous value after every $50$ epochs. For Stage-I, we use a batch size of $384$ and for Stage-II, we use a batch size of $64$. In case of the recurrent dialogue encoder, the hidden dimension of the RNN is set to $1024$. The implementation is based on *PyTorch* (Paszke et al., 2017) and we trained the models on a machine with 4 NVIDIA Tesla P40s.

## C  EVALUATION

For computing inception score, we use the Inception v3 model pretrained on ImageNet available with *PyTorch*. We then generate images for the 40k test set and use 10 random splits of 30k images each. We report the mean and standard deviation across these splits.

## D  RESULTS: EXTENDED

We present some of the more realistic images generated by our non recurrent encoder ChatPainter in Figure 3 and by our recurrent encoder ChatPainter in Figure 4. For fairness of comparison, we

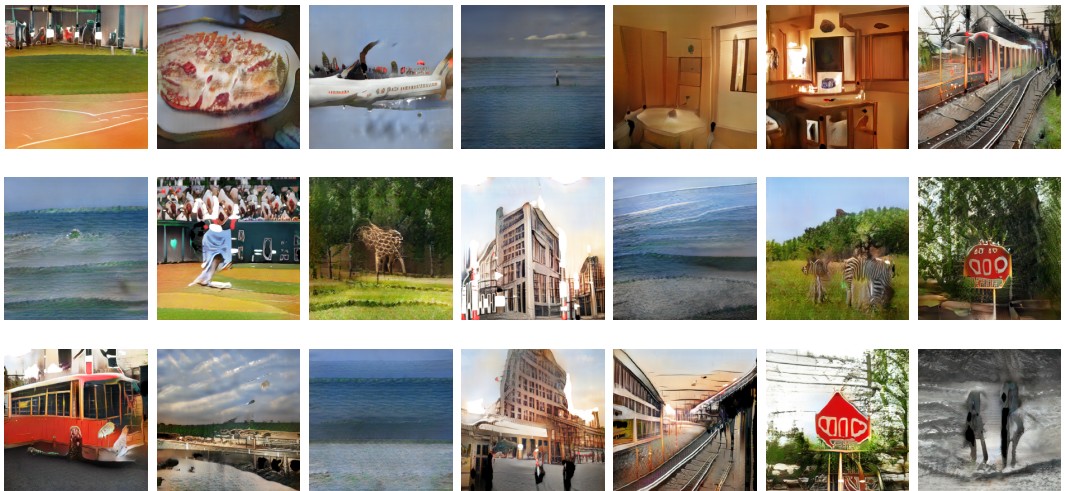

Figure 3: Example $256 \times 256$ images generated by our non-recurrent encoder ChatPainter model on the COCO test set. Best viewed in color. Images are cherry-picked from a larger random sample.

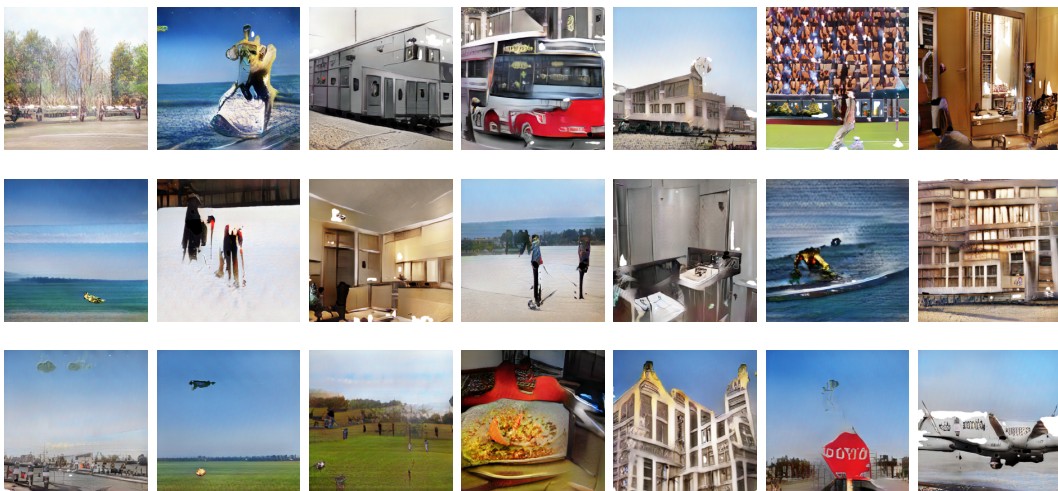

Figure 4: Example $256 \times 256$ images generated by our recurrent encoder ChatPainter model on the COCO test set. Best viewed in color. Images are cherry-picked from a larger random sample.

also present a random sample of the images generated by our recurrent encoder ChatPainter on the COCO dataset in Figure 5. As seen from these figures, the model is able to generate close-to-realistic images for some of the caption and dialogue inputs though not very realistic ones for most.

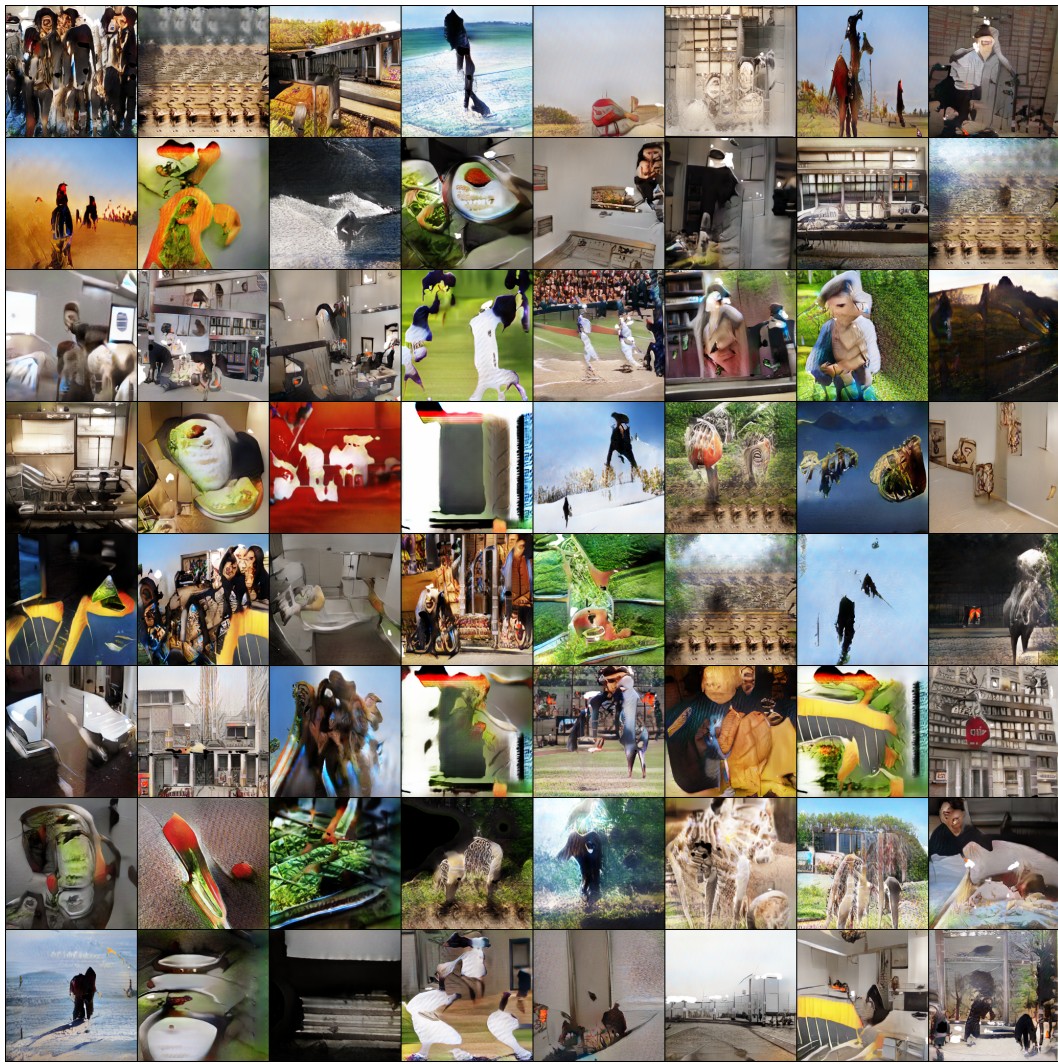

Figure 5: Example $256 \times 256$ images generated by our recurrent encoder ChatPainter model on the COCO test set. Best viewed in color. Images have been selected randomly.

