# OpenReview forum: "ChatPainter: Improving Text to Image Generation using Dialogue"
_ICLR.cc/2018/Workshop — Accept_

### Official Review · AnonReviewer1 · 2018-02-25
**An interesting idea**

**Rating:** 5
**Confidence:** 4

**Review:**

The paper proposes to improve automatic generation of images from textual description, by using a "dialog" that describes the image rather than only using a textual description.  The proposed model is based on StackGAN

Strengths: The problem is interesting, timely and important, and the idea of using a more interactive form of image description is promising.

Weakness: The paper does not provide evidence that using a dialog helps, and it is not clear why a description formatted as questions and answers would be more useful than a detailed caption. The most relevant experimental result are given in Appendix Table 2, showing a dialog and an image, but the effect of the various parts of the text dialog are very unclear. The rest of the experiments show generated images, but we done have their captions or dialogs so it is not possible to evaluate the contribution of the text in creating these images.

Comment: There is interesting related work by Justin Johnson1 (still unpublished).

---

### Official Review · AnonReviewer2 · 2018-03-10
**First and interesting result on image synthesize conditioned on dialog.**

**Rating:** 6
**Confidence:** 4

**Review:**

This paper proposed to synthesize the realistic image from the dialog instead of image captions. (visDial Dataset).  The authors use 2 different encoders (Non-recurrent and recurrent) to extract the dialog features and adapt stack GAN to synthesize the image. Inception score verifies that chatpainter which conditioned on the dialog data is better than simple conditioned on captions. The authors also provide a lot of insights in the discussion and future work section. Image generation from interaction conversation is a very interesting problem and application.

Cons:
I would expect the authors to show more qualitative results with vary dialog history length, instead of the final generated images. I'm curious how the dialog length affects the final image generation result.

---

### Decision · Program_Chairs · 2018-03-20
**ICLR 2018 Workshop Acceptance Decision**

**Decision:**

Accept

**Comment:**

Congratulations, your paper was accepted to the ICLR workshop.